# Confidence interval forecasting model of small watershed flood based on compound recurrent neural networks and Bayesian

**Songsong Wang**[1]*, Ouguan XU[2]

**1** Nanxun Innovation Institute, Zhejiang University of Water Resources and Electric Power, Hangzhou, China, **2** School of Computer Science and Technology, Zhejiang University of Water Resources and Electric Power, Hangzhou, China

\* wss@zjweu.edu.cn

## Abstract

Flood forecasting exhibits rapid fluctuations, water level forecasting shows great uncertainty and inaccuracy in small watersheds, and the reliability and accuracy performance of traditional probability forecasting is often unbalanced. This study combined Recurrent Neural Networks (RNN) and Bayesian to establish a comprehensive forecasting model framework of RNNs-Bayesian for the forecasting of water level confidence interval, to achieve both reasonable reliability and accuracy. In the Bayesian structure, weight training was used. In the RNNs, base RNN, Long Short-term Memory (LSTM), and Gated Recurrent Unit (GRU) are used for comparative analysis, and experiments are carried out at the point of the Qixi Reservoir in a small watershed in Zhejiang Province of China. We used the multidimensional disaster data input unit for water level forecasting, including hydrology, meteorology, and geography, and 5 days of time windows for forecasting, The comprehensive reliability of LSTM-Bayesian for 0~102 hours flood reached 92.31%, and the comprehensive accuracy reached 89.15%, and confidence interval forecasting using LSTM is the best method, and achieved reasonable balance of reliability and accuracy. Overall, compound RNN could be a good alternative for forecasting hourly streamflow and extreme water level in small watersheds.

## 1 Introduction

Flood in small watersheds is mainly influenced by uncertain natural factors, including hydrology, meteorology, geography, etc., the usual forecasting of water level methods are mainly deterministic values, which lack the balance between reliability and accuracy. At present, the main approach in hydrology is based on physical models of hydrology and hydrodynamics, with deterministic forecasting as the main approach and a lack of methods for uncertain forecasting. The error rate of flood warning is high, it is of great significance to study the prediction of flood in small watersheds to reduce the complex sudden flood disasters. With the research of Artificial Intelligence (AI) into uncertain forecasting, AI can effectively improve forecasting techniques based on industry physical models.

**Data availability statement:** The data underlying the results presented in the study are available from https://slt.zj.gov.cn/ and https://data.cma.cn

**Funding:** This research funded by Joint Funds of the Zhejiang Provincial Natural Science Foundation of China under Grant (Grant No. LZJWZ23E090001), and Nanxun scholars program of ZJWEU (RC2025010997), these fundings were jointly received by Songsong Wang(the first author) and Ouguan Xu.

**Competing interests:** The authors have declared that no competing interests exist.

We aim to establish an uncertainty forecasting methodology for the balance of reliability and accuracy, utilizing advanced Bayesian and Recurrent Neural Networks (RNN), and data mining, with the objective of attaining high reliability and accuracy in forecasting. In the RNNs, base RNN, Long Short-term Memory (LSTM), and Gated Recurrent Unit (GRU) are used for certainty forecasting. The study examines an uncertainty forecasting model grounded in hydrological physical mechanisms. We propose RNNs-Bayesian forecasting models that are adept at translating spatiotemporal factors associated with floods into disaster probabilities. The models under consideration include base RNN-Bayesian, LSTM-Bayesian, and GRU-Bayesian, with the intention of achieving a satisfactory equilibrium between accuracy and reliability. Additionally, we aim to extract pertinent information from a diverse array of multi-source and multi-dimensional hazard factors' Big Data (BD), which is anticipated to enhance the interpretability of the forecasting outcomes.

Through an examination of the uncertainty theory pertaining to forecasting and testing, it has been determined that the forecasting capabilities of the proposed model remain within the bounds of the reliability forecasting interval, even during periods characterized by rapid fluctuations in water levels. This model achieves a commendable equilibrium between reliability and accuracy. The approach employed involves the utilization of a multidimensional disaster data input unit for water level forecasting, with LSTM networks identified as the most effective method for confidence interval forecasting. Additionally, we have employed various integrated regression of Machine Learning (ML) in conjunction with Bayesian methods to ascertain the optimal combination for forecasting. A significant contribution of this research lies in the identification of a methodology that effectively merges time series accuracy forecasting with probabilistic assessments within the context of small watershed flood forecasting. By integrating RNNs with Bayesian principles, we have developed a confidence interval forecasting model that enhances both the reliability and accuracy of flood forecasting. It improves the reliability of flood warning in small watersheds, and can provide reliable information for the decision-making of flood control and disaster management of government departments to avoid flood disasters.

## 2  Related works

Deep Learning (DL) has emerged as a sophisticated AI approach for time series forecasting [1], particularly exemplified by the utilization of multi-dimensional and multi-step data, DL and Bayesian probability models. In the domain of hydrology forecasting, challenges arise from the presence of complex and high-dimensional data sources, which complicate the identification of critical information that may lead to disasters [2,3]. However, the implementation of advanced BD technologies for effective information extraction in water conservancy can significantly improve the reliability of DL methodologies. Concurrently, there remains a need to enhance the theoretical interpretability of DL techniques grounded in data intelligence [4]. Consequently, investigating uncertainty DL methods of RNN that leverage BD related to flood disaster factors represents a vital avenue for accurately forecasting flood water levels [5,6].

In terms of deterministic forecasting, RNNs are the key technology for deterministic forecasting. At its core, the RNNs can learn all the hidden states within the time series before forecasting as feature representations of past information, and combine the current input to provide the next forecasting result. At each time step, new observations can be used to continuously recursively update the hidden layer state. Therefore, the base RNN was also the earliest applied in time forecasting scenarios in DL methods. Early base RNN variants may be limited by gradient explosion and vanishing problems when learning long-term serial dependencies in data [7].

One solution is to use LSTM networks [8]. The design inspiration for long-term and short-term memory networks comes from the logic control gates of computers, by using various gates to control memory elements, such as using output gates to control the output sequence from the unit, by using input gates to determine when to read data into the unit, by using forget gates to manage the content of reset units. The weights obtained through training in LSTM can determine when to remember or ignore inputs in hidden states. Therefore, LSTM and its variant GRU units have become important components of base RNN for time series forecasting [9,10].

In terms of uncertainty forecasting, the main technology for the development of probability intervals is based on Bayesian theory [11]. Bayesian of flood forecasting method is an advanced and effective method for estimating flood probability. Bayesian combined with physical model can improve interpretability. Multiple hydrological model comparison and an improved Bayesian model averaging approach for ensemble forecasting over semi-humid regions [12], and urban flood numerical simulation [13]. In addition, the DL method is gradually integrated, such as multi-step ahead runoff forecasting [14], and hybrid DL applications for streamflow forecasting [15]. Probabilistic forecasting begins to integrate with other algorithms, such as temporal Convolutional Neural Network (CNN) [16], and integrated model chain for future flood risk forecasting [17]. In short, Bayesian is gradually integrating DL into uncertainty forecasting to achieve better comprehensive results.

The combination of deterministic forecasting and probabilistic forecasting can give consideration to the accuracy and reliability of forecasting. Uncertainty model is the extension of hydrological models, by using Bayesian and LSTM with stochastic variational inference for estimating [18,19], and residual errors of LSTM [20]. They can use for estimating uncertainty and data analysis [21]. The improved dynamic Bayesian networks structure [22] and Bayesian model averaging approach improved the reliability and accuracy of hydrological forecasting [12,23]. The data-driven model for deterministic forecasting by DL is feasible and efficient.

In addition, Bayesian based uncertainty forecasting is still facing great challenges of complex predicting. The complex fusion structure of Bayesian and DL is not easy to design, to ensure that combining Bayesian methods with DL enables to express uncertain status in DL models [24,25]. The generalized performance of the data-driven models during training [26]. Modeling uncertainty has been a major challenge with developing Machine Learning (ML) solutions to solve real-world problems in various domains.

In the evaluation of uncertainty forecasting, the key confidence interval forecasting evaluation is generally aimed at the forecasting interval with a certain specified confidence level, such as the 90% confidence interval. The commonly used confidence interval forecasting accuracy evaluation indicators include Coverage Rate (CR), average bandwidth and average Relative Bandwidth (RB), average offset amplitude, etc. The closer the coverage CR value is to the specified confidence level, the better [27,28]. For a specified confidence level, the narrower the average RB in the forecasting interval, while ensuring high coverage, is better. In theory, the smaller the average deviation amplitude, the better the symmetry of the forecast interval [29,30].

It cannot be ignored that data also plays an important role in AI driven forecasting. Data is the determinant of RNNs-Bayesian model. The data-driven method is closer to the physical principle [31,32]. The combination of data estimation and ML improves the forecasting ability [33]. In water conservancy applications, accurately predicting water levels is a challenge due to the non-stationarity of hydrological processes and the influence of noise. A data-driven model by using a dual data processing strategy, which can improve forecasting accuracy through denoising processing [34]. The advanced data-driven model can be more effective when combined with the proposed preprocessing method [35]. RNNs and multi-dimensional data

can significantly improve the forecasting level of flood water level [36]. In a word, data is the basis. Through in-depth information extraction of multidimensional BD of impact factors, the performance of forecasting methods can be improved.

In conclusion, the theoretical framework for forecasting floods in small watersheds requires enhancement to increase its interpretability, as the uncertainty associated with data on factors contributing to disasters significantly influences forecasting outcomes. Employing probabilistic methods to represent uncertainty within the forecasting domain can yield high reliability. Furthermore, utilizing RNNs for forecasting offers timeliness, accuracy, and ease of implementation.

## 3 Methodology

### 3.1 Uncertainty theory of forecasting

In terms of water level forecasting, both hydrodynamic models and RNNs have a common forecasting expression (1). $X$ is the time series input variable (impact factors), $Y$ is the predicted value series, and $\theta$ is the model parameter.

$$Y = f\left(X, \theta\right) \tag{1}$$

We use RNNs model to approximate the physical forecasting model, use Kullback-Leibler (**KL**) divergence [37] to measure the distance between these two distributions, achieving approximate variation, i.e., optimization:

$$\boldsymbol{KL}(\boldsymbol{q}\,||\,\boldsymbol{p}) = \sum_{i=1}^{n} \boldsymbol{q}\left(\boldsymbol{\theta}\right)\log\frac{\boldsymbol{q}\left(\boldsymbol{\theta}\right)}{\boldsymbol{p}(\boldsymbol{\theta}\,|\,X,Y)} \tag{2}$$

Variational estimation, as outlined in equation (4), involves the dynamic adjustment of variables $X$, $Y$, and parameters $\theta$ to enhance the forecasting capabilities of BD and AI. This process entails the dynamic modification of $X$ and $Y$ through the effective information extraction techniques developed by the applicant, which are designed to address new multi-source and multi-dimensional uncertainties in forecasting. Additionally, the parameters are dynamically adjusted through the innovative construction of new RNNs forecasting models represented by $\theta$. It is anticipated that these methodologies will lead to a substantial improvement in predictive performance.

Regarding the parameter θ, an approximate variation of the Gaussian distribution is applied to equation (2). Consequently, the weight and deviation distribution at the $n_{\text{th}}$ layer of the model for the $i_{\text{th}}$ time step is expressed as follows:

$$W_{(n)}^{(i)} = \mathcal{N}\left(0,1\right) * \log\left(1 + \rho_W^{(i)}\right) + \mu_W^{(i)} \tag{3}$$

$$b_{(n)}^{(i)} = \mathcal{N}\left(0,1\right) * \log\left(1 + \rho_b^{(i)}\right) + \mu_b^{(i)} \tag{4}$$

Further, the uncertain expected data such as probabilistic rainfall in the weather forecast are integrated, and the disaster a priori probability derived from the historical data of flood disasters is used to construct a DL Bayesian network probabilistic longitudinal forecasting time disaster model and horizontal time series disaster model, set $X^{(t_i)} = \left\{M_1^{t_i}, M_2^{t_i}, \cdots, M_n^{t_i}\right\}$, represents the set of n influencing factor variables in the time slice $t_i$, then the distribution of the joint probability is:

$$P\left(X^{(t_0)}, X^{(t_1)}, \cdots, X^{(t_m)}\right) = \prod P\left(X_i | P_a\left(X_i\right)\right) \qquad (8)$$

Let the prior distribution of the parameter be $P\left(w_1, w_2, \cdots, w_n\right)$, given a set of training examples $D = (x_i, y_i)$, when we use RNNs dynamic Bayesian network to predict, the relationship between the constantly updated input value and the predicted value, the relationship between the continuously updated input value and the predicted value when we use a dynamic Bayesian network for forecasting:

$$\hat{y}_{N+1} = \int f\left(x_{N+1}, w\right) P(w|D) dw \qquad (9)$$

On the premise that the predicted model structure and node parameters are determined, the conditional probability formula can predict the Bayesian probability distribution of water level and disaster level at a certain time from 1 to 7 days and 7–30 days.

### 3.2 RNNs-Bayesian model

With the permission of Zhejiang University of Water Resources and Electric Power, we construct a deterministic and uncertain composite forecasting model based on the uncertainty forecasting theory. We design the RNNs framework based on multiple technologies including base RNN, LSTM and GRU to train and continuously correct composite models, a deterministic RNNs model scheme for each forecasting point is constructed. The dynamic Bayesian method is used to construct an uncertainty probability RNNs forecasting supplementary model based on historical prior information and parameter periods, as shown in Fig 1.

In the multi-source data block, the historical data time series set and real-time data stream are selected, divided into training set and test set, and the real-time data stream is transformed into forecasting's input data. The input data need standardized processing, data segmentation for RNNs model and Bayesian block for uncertainty forecasting. RNNs blocks include the model training and continuous correction. Among them, $X$ is the input; cells include the hidden multi-layer RNNs units; $P$ is the actual output; $C_h$ are the input and forget gate parameters; $L$ is the number of cells. The training process includes the theoretical output, loss calculation and actual output. The model achieves optimal parameters for each layer through repeated training, test and prediction, and the use of the Adam error training method. Bayesian block's main function of is training for Bayesian parameter estimation, including weight estimation ($W$) and Bias estimation ($b$). Finally, the result block of Water level/ hazard level forecasting and measurement time series set, a forecasting model library is constructed to predict multiple small watersheds for Short-Term (0~24 hours), Mid-Term (1~7 days) and Long-Tern (1~30 days)'s eater level and disaster level.

The structure of the RNNs are shown in Fig 2, the RNNs' equations in RNNs-Bayesian model is shown in Table 1, and the parameter's interpretation is shown in Table 2.

Base RNN is a feedforward neural network with temporal connections, they have states, channels and temporal connections between them. The input information of a neuron includes not only the output of the previous neuron layer, but also its own state in the previous channel. The LSTM method can preserve information over long periods through its unique structure, cell state, and gates to control how the information flows in the different layers.

In order to solve the computing power of the training model, GRU can reduce computational complexity, the speed of training has been improved. Similar to LSTM, it solves the problems of long-term memory and gradients in backpropagation in RNN through gating units. Comparing to LSTM, the internal network architecture of GRU is relatively simple. The GRU network contains two gates that use an update gate and a reset gate. The reset gate

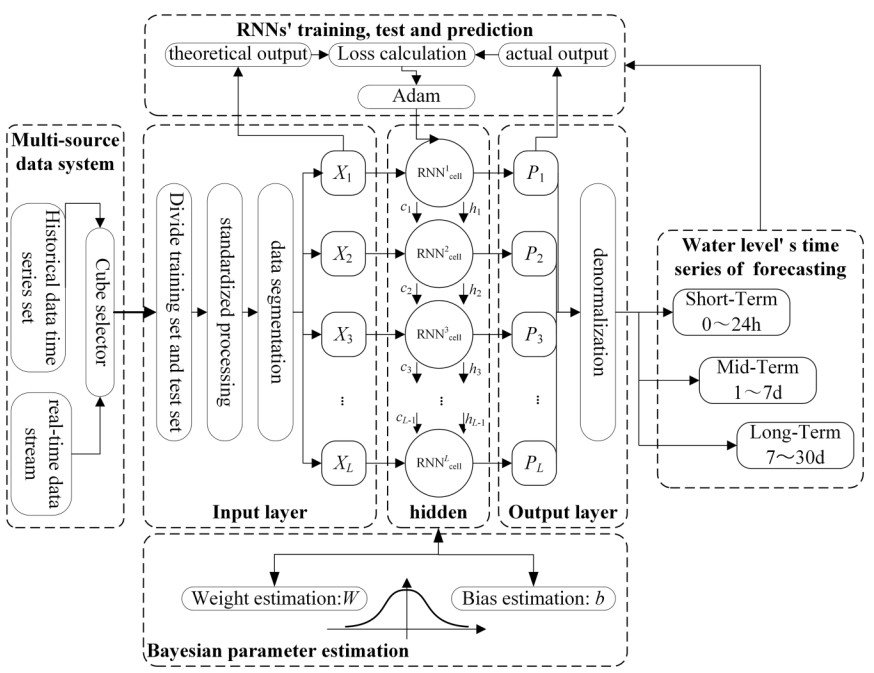

**Fig 1. The entire architecture of RNNs-Bayesian model.**

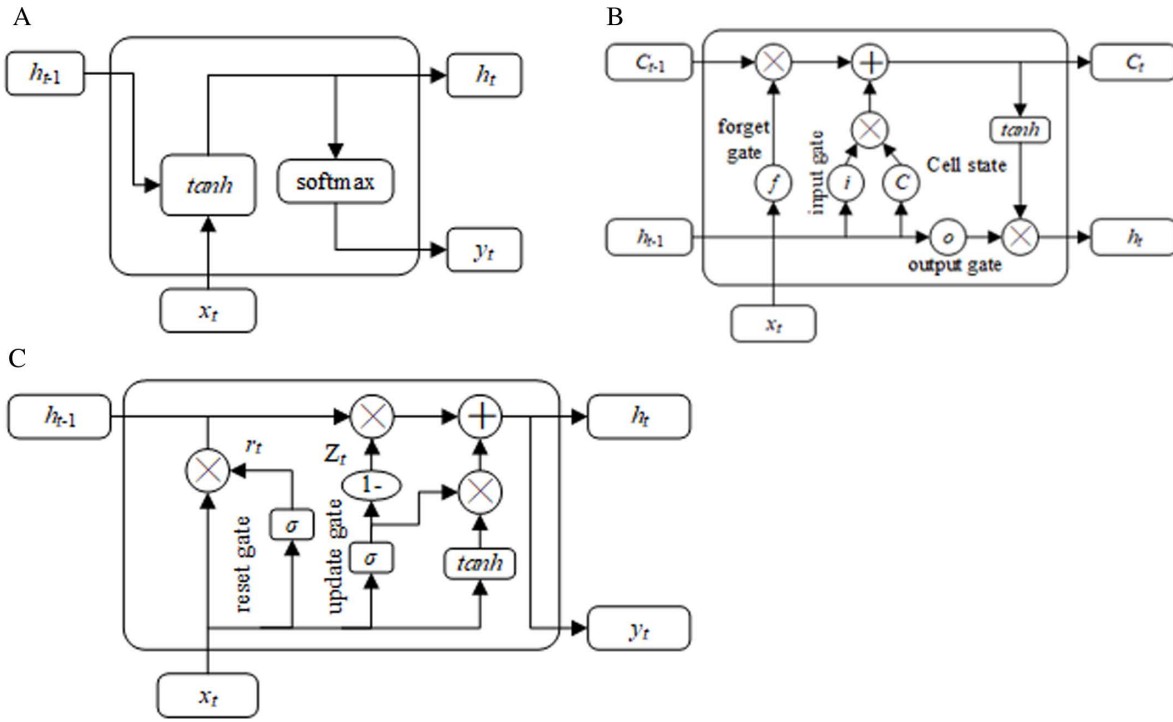

**Fig 2. The architecture of RNNs in RNNs-Bayesian model.** (a) Base RNN (b) LSTM (c) GRU.

**Table 1. The RNNs' equations in RNNs-Bayesian model.**

| Name | Base RNN-Bayesian | LSTM-Bayesian | GRU-Bayesian |
|---|---|---|---|
| Input | $x_t$ | $i_t = \sigma\left(W_i x_t + U_i h_{t-1} + b_i\right)$ | $\widetilde{x_t} = W x_t$ |
| Forget/ update gate | / | $f_t = \sigma\left(W_f x_t + U_f h_{t-1} + b_f\right)$ | $f_t = \sigma\left(W_f x_t + b_f\right)$ |
| Reset gate | / | / | $r_t = \sigma\left(W_r x_t + b_r\right)$ |
| Cell input | / | $\widetilde{C_t} = \tanh\left(W_c x_t + U_c h_{t-1} + b_c\right)$ | / |
| Cell state | / | $C_t = f_t \otimes C_{t-1} + i_t \otimes \widetilde{C_t}$ | $C_t = f_t \otimes C_{t-1} + \left(1 - f_t\right) \otimes \widetilde{x_t}$ |
| Hidden state | $h_t = \tanh\left(x_t W_{xh} + h_{t-1} W_{hh} + b_h\right)$ | $h_t = O_t \otimes \tanh\left(C_t\right)$ | $h_t = r_t \otimes g\left(C_t\right) + \left(1 - r_t\right) \otimes x_t$ |
| Output (gate) | $y_t = \text{softmax}\left(h_t W_{hy} + b_y\right)$ | $O_t = \sigma\left(W_O x_t + U_O h_{t-1} + b_O\right)$ | $h_t$ |

**Table 2. Parameter's interpretation.**

| Param | Interpretation |
|---|---|
| $x_i$ | The input data, including hydrological, meteorological, geographic, and crowdsourced data on the same time scale. |
| $I_t$ | The input gate |
| $F_t$ | The forget gate |
| $O_t$ | The output gate |
| $W_i$ | The weight of connecting input data and the input gate |
| $W_f$ | The weight of connecting input data and the forget gate |
| $W_o$ | The weight of connecting input data and the output gate |
| $W_r$ | The weight of connecting input data and the reset gate |
| $U_i$ | The weight of connecting input gate and the hidden state |
| $U_f$ | The weight of connecting forget gate and the hidden state |
| $U_o$ | The weight of connecting output gate and the hidden state |
| $B_i$ | The input gate bias vector |
| $B_f$ | The Forget gate bias vector |
| $B_o$ | The output gate bias vector |
| $C_{t-1}$ | The cell input |
| $C_t$ | The current cell state |
| $h_t$ | The current hidden state |
| $\Sigma$ | The logistic sigmoidal function |
| $\otimes$ | The element-wise multiplication |
| $g$ | The activation function |
| Tanh/ softmax | The hyperbolic tangent function/ normalized exponential function |

determines how to combine new input information with previous memory, while the update gate defines the amount of previous memory saved to the current time step.

## 4 Data and results

### 4.1 Data

The upper reaches of the Qiantang River are characterized by mountainous terrain, with the Qixi Reservoir monitoring point located in Kaihua County, Zhejiang Province, China, which

marks the source of the Qiantang River. Kaihua County experiences a subtropical monsoon climate, situated on the northern edge of this climatic zone. This region is known for its warmth, humidity, abundant rainfall, and distinct four seasons. The annual average precipitation in Kaihua County is 1990 mm, ranking second highest in Zhejiang Province.

Rainfall in the Qiantang River watershed is primarily concentrated during the summer months (June to September), particularly during the plum rain and typhoon seasons. Heavy rainfall during these periods is the primary cause of floods in the region. The upper reaches of the Qiantang River feature steep terrain, narrow river valleys, and rapid water flow, conditions that facilitate the rapid formation of floods. Floods often develop within a short time frame, typically hours or even minutes after heavy rainfall, posing significant challenges for early warning systems and emergency response efforts.

We collected hourly data for 4 years (2021~2024) of typical flood monitoring point of Qixi Reservoir, including weather, rainfall, temperature, humidity, wind direction, wind speed and other meteorological data; rainfall, water level and other hydrological data. The hydrological data mainly come from the water conservancy management department of Zhejiang Province (https://slt.zj.gov.cn/), and other data come from China Meteorological Data Service Centre (https://data.cma.cn).

We establish a connection to a multi-source data platform that encompasses key influencing factors, including hydrology (rainfall, water level), meteorology (weather status, wind, temperature, humidity, etc.), and geography. This platform is integrated with the research and development team's water conservancy database to create a comprehensive data center aimed at facilitating the large-scale collection of historical, real-time, and forecast data. Following the processes of data preprocessing, fusion, and storage, we prepare the data for use in a compound forecasting model, with model parameters stored in a dedicated model database to support the development of an automated testing environment.

Furthermore, we employ advanced instruments such as terrain scanning drones and multi-wave speed river survey boats to gather high-density geographic data within the test area. A mobile Internet of Things (IoT) device for water conservancy is also established at the test site to collect real-time hydrometeorological and geographic data, thereby enhancing the timeliness and accuracy of the model's input data. Our analysis utilizes data from 12,000 consecutive hours, spanning the flood period from March 2023 to July 2024, with the primary rainfall and water levels illustrated in Fig 3.

The periodicity and regularity of fluctuations in water levels are not readily apparent, rendering traditional regression-based forecasting methods ineffective. A comprehensive

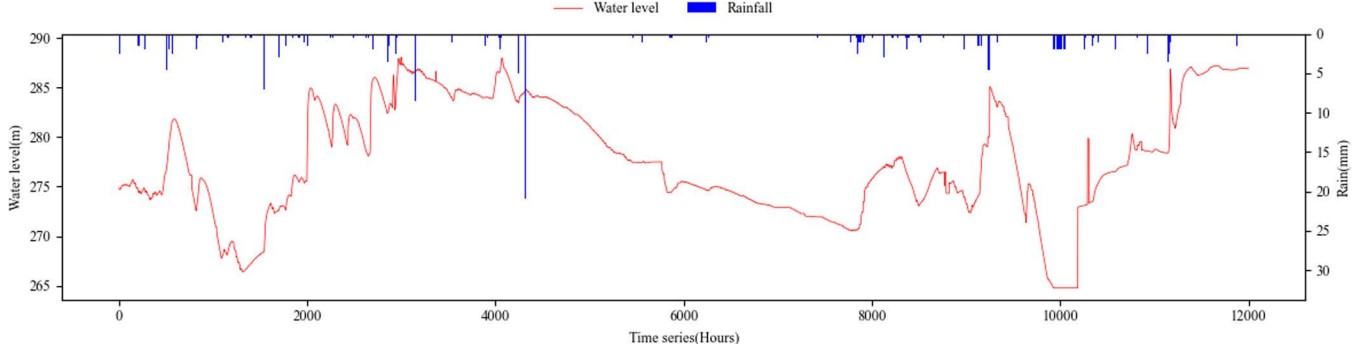

**Fig 3. Study Area's main data (water level and rainfall).**

examination of the input data is essential. The characteristics of the input data can be summarized as follows.

(1) There exists significant uncertainty associated with the factors contributing to disasters, necessitating the enhancement of data reliability through techniques such as data imputation, cross-validation, and peer review.

(2) Both rainfall and water level data exhibit a normal distribution, and the water level demonstrates a degree of periodicity, making it amenable to Bayesian probability forecasting.

(3) Although the data is collected at one-hour intervals, the overall trend of the predictive data changes is relatively smooth on a macro scale. However, the high variability and rapid fluctuations observed on a micro scale may compromise the continuity and precision of the forecasting probabilities.

There are many disaster-causing factors affecting future water level prediction, including hydrology, meteorology and geography, etc. In order to obtain the factors with great influence, we use the relationship matrix, add the relationship matrix to the analysis of the relationship between the flood disaster-causing factors and the water level, as shown in Fig 4.

The future water level to be predicted is mainly related to the historical water level, up to 58%, and the relationship with rainfall is up to 28%, which is consistent with the basic law of hydrology. In addition, humidity, weather status, geography, etc. are also closely related to the future water level, and these data will be used as the input elements of the uncertainty prediction model.

## 4.2 Experiment environment

Python is a programming language that can provide RNNs models, and with PyTorch library are efficient tools for predictive data analysis. The extensible library to create Bayesian neural

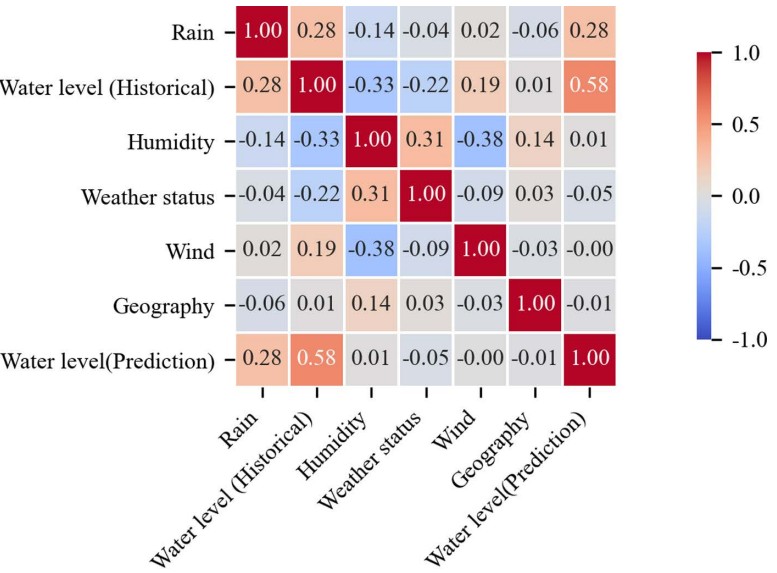

**Fig 4. Relationship matrix between main disaster-causing factors and water level of flood.**

**Table 3. The models' training parameters.**

| Parameter | Value | Meaning |
|---|---|---|
| units | 64 | Positive integer, dimensionality of the output space. |
| Learning Rate | Adam (Adaptive Learning Rate) | The proportion of inputs that need to be discarded. |
| activation | relu | Activation function |
| kernel_ regularizer | tf.keras.regular-izers.l2() | Regularizer function applied to the kernel weights matrix. |
| adam | 0.0001 | Adam optimization function and value. |
| loss | mean_squared_ error | Loss function |
| Metrics | accuracy | List of metrics to be evaluated by the model during training and testing. |
| Batch_size | 8 | Integer or None. Number of samples per batch of computation. |
| Epochs | 11000 | Number of epochs to train the model. |
| Time window | 120 hours | The data of flood water level at the future time point (each time is the next hour) is predicted through the data of disaster causing factors in the moving time window. |

network layers on PyTorch at GitHub (https://github.com/) is used. Calculations are carried out with the CPU of intel(R) Xeon(R) E-2176M, double cores, 2.70GHz and 32 GB RAM.

The ensemble RNN models include base RNN-Bayesian, LSTM Bayesian and GRU Bayesian, and the three models of RNNs-Bayesian have the similar structures, the only difference is the type of data RNN layer, one is a RNN mechanism and the other is an LSTM structure. The models have the main parameters' value and meaning at Table 3. The parameters are set as the best result obtained through adaptive calculation based on the model.

To ensure comparability among different models, the modeling boundary conditions were maintained as consistently as possible. This involved keeping the input data, the number of layers, the number of neurons, the dropout rate, and the number of training epochs identical. As a result, each model developed a unique set of parameters while retaining the same structure. Following the training process, the values were denormalized to facilitate comparison with observed data.

In this section, the comparative performance analysis of Bayesian methods for long-term forecasting is presented. Compared with the uncertainty models by using various statistics measure, as shown as Table 4.

**Table 4. The Evaluation parameters of forecasting results.**

| Parameter | Full name | Meaning |
|---|---|---|
| PICP | Prediction Interval Cover-age Probability | It is the ratio of the point where the actual value falls within the confidence inter-val to the actual value, which means the interval coverage probability of Bayesian RNNs' output. The PICP is higher, the higher the reliability of the forecasting. |
| CIAW | Confidence Interval Aver-age Width | It is the average width of a specific forecasting interval, the smaller its value, the higher the accuracy. |
| MDAD | Median Devi-ation Average Difference | It is the average difference between the median value of the prediction probabil-ity interval and the actual value. The smaller it is, the better the comprehensive performance of reliability and accuracy is. |
| / | Reliability | The average value of comprehensive PICP in test and prediction phases' is in the range of [0,1]. The larger the value, the more accurate it is. |
| / | Accuracy | For the ratio of the average value of CIAW in test and prediction phases', in the [0,1] interval, the greater the value, the more accurate it is. |

We use base RNN-Bayesian, LSTM-Bayesian and GRU-Bayesian for comparative analysis, mainly focusing on PICP (i.e., reliability) and CIAW (i.e., accuracy). However, PICP and CIAW are a pair of contradictory results:

(1) If increase the width of the confidence interval, then it should improve reliability, but reduces accuracy.

(2) If reduce the confidence interval width, then is should reduce reliability, but improves accuracy.

The experiment includes the training, test and prediction phase, the test is adjusted by the real water level for training, but prediction phase doesn't have it, and simulate the real prediction. The three phases respectively account for 80%, 10%, and 10% of the entire time series data, we gradually improve the effectiveness of forecasting through these three-tiered phases. We hope to achieve an acceptable balance between PICP and CIAW, the larger the probability value of its confidence interval, the more reliable it is, and the narrower the confidence width, the more accurate it is. The smaller of MDAD, the better the comprehensive performance of reliability and accuracy is.

## 4.3 Results

**Training epochs.** Fig 5 shows the training error situation, Table 5 shows the evaluation of RNNs-Bayesian models' training. In the training phase, the training errors of each algorithm can quickly converge to stability within 300 steps. The GRU-Bayesian is the fastest and takes the shortest time of 43 minutes. However, when the initial training error drops rapidly, it has certain instability. The training error reduction of base RNN-Bayesian belongs to the process of hard landing, and the error does not rebound greatly, but the error reduction has a certain delay. It can be gradually accepted by the algorithm after 2000 steps, and the training time is the longest. LSTM-Bayesian has the best stability and speed of error reduction. In general, the training errors of the three models are convergent and stable, which proves the feasibility of the model in this paper.

Figs 6–8 show the overall effect of base RNN-Bayesian, LSTM-Bayesian and GRU-Bayesian, as well as the details of the prediction phase, respectively, the Figure (a) shows the Train, test and prediction phases' Overall graph, the Figure (b) shows the test and prediction phases' detail graph. Table 6 explained the meaning, range, and color of each data of the label in the figures.

Table 7 shows the results of RNNs-Bayesian in the test and prediction phase. In the test phase, the PICP of base RNN-Bayesian did not reach 70%, and the reliability was insufficient. The LSTM-Bayesian and GRU-Bayesian reached 100%, indicating that the model capability of LSTM-Bayesian and GRU-Bayesian sufficient in the test phase. Although the PICP of LSTM-Bayesian reached 100%, its CIAW was too large, indicating that LSTM-Bayesian increased the coverage of confidence interval to ensure reliability. For MDAD elements, it is obvious from the renderings of the three models that the coincidence degree between LSTM-Bayesian and the actual water level is the highest, the average difference is 0.01m, and the lowest, while base RNN-Bayesian and GRU-Bayesian have obvious deviation, which shows that LSTM-Bayesian takes into account the reliability and accuracy in the test phase, and the comprehensive effect is the best.

In the key phase of prediction, LSTM-Bayesian still maintained the advantages of the test phase, especially in the 0-72h phase, the PICP reached 96.18%, which was the closest to the actual water level, and the MDAD was only 0.01m, which had a good prediction effect, taking into account the reliability and accuracy. GRU-Bayesian also performed well, the only

disadvantage is that MDAD still exists large error, and 10 times less than LSTM-Bayesian, but it is also within the acceptable range. The deviation of base RNN-Bayesian is the weakest among the three. Another obvious feature is that when the water level rises rapidly, the prediction effect is better, and the main disaster causing factors, mainly rainfall, are considered in the model. In the 72-102h phase, LSTM-Bayesian and GRU-Bayesian had a large deviation, and PICP could not cover the actual water level. base RNN-Bayesian covered the actual water level due to PICP with a large coverage rate, and the MDAD value was relatively stable. At the same time, the applicability of water level prediction was reduced due to too large PICP. In the prediction phase, 0-72h maintained the advantage, and the three predictions were effective.

To facilitate statistics and analysis, we convert PICP and CIAW into comprehensive reliability and accuracy. The reliability of base RNN-Bayesian and LSTM-Bayesian models is higher than GRU-Bayesian model, reaching 84.34% and 92.31%, respectively. The accuracy of LSTM Bayesian and GRU-Bayesian prediction models is higher than base RNN-Bayesian model, reaching 89.15% and 93.05%, respectively. Taken together, LSTM-Bayesian model has advantages in reliability and accuracy.

On the whole, the experiments have proved that the three models are effective for water level forecasting. LSTM-Bayesian has the best comprehensive effect in terms of reliability, accuracy and operation speed. GRU-Bayesian has the best effect in terms of reliability and operation speed. base RNN-Bayesian has better training effect, but it is not as good as the other two in the test and prediction phases. All RNNs' reliability and accuracy of long-term prediction are very low.

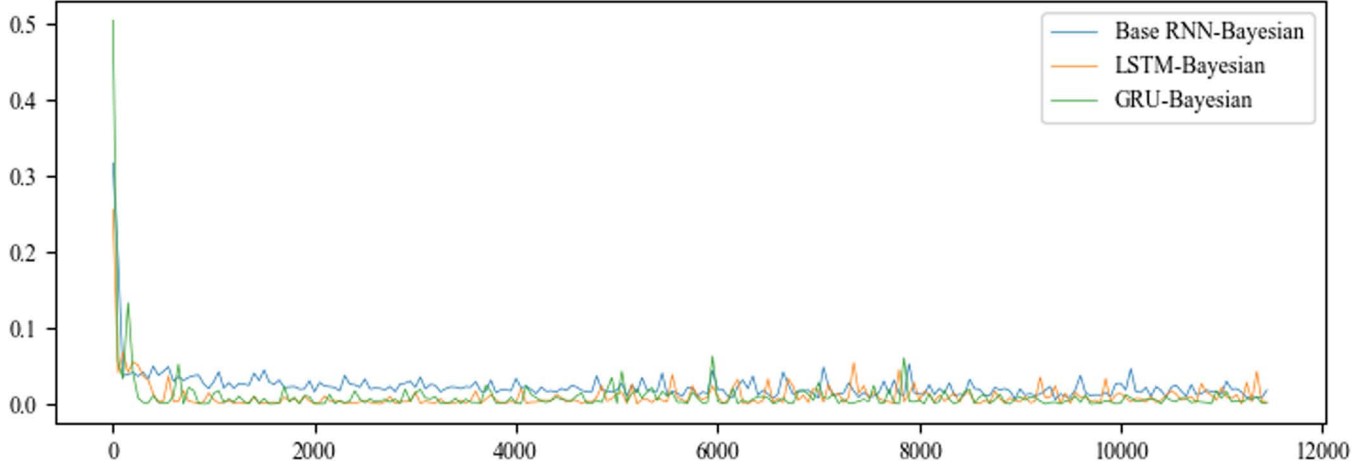

**Fig 5. RNNs-Bayesian models' training error.**

**Table 5. The evaluation of RNNs-Bayesian models' training.**

| No | Type | Meaning | Base RNN-Bayesian | LSTM-Bayesian | GRU-Bayesian |
|----|------|---------|-------------------|---------------|--------------|
| 1 | Computing time of training | Computing time for training by minutes. | 53 | 49 | 43 |
| 2 | Steps of training | Computing steps of training when the error is stable. | 225 | 153 | 83 |

 

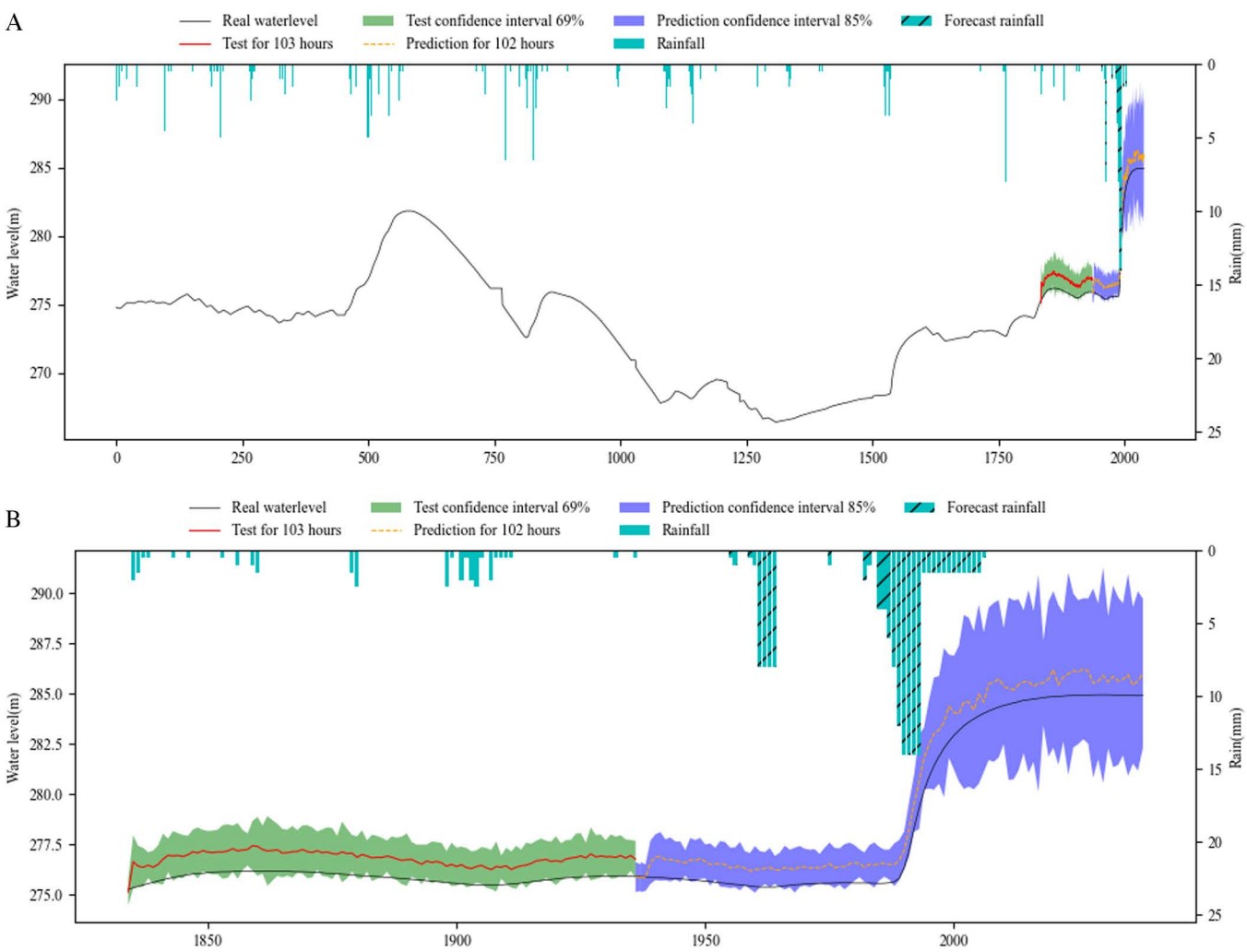

**Fig 6. Phases of base RNN-Bayesian model's test and prediction.** (a) Training, test and prediction phases' overall graph. (b) Test and prediction phases' detail graph.

## 5 Discussions

### 5.1 Comparison of forecasting models

At present, the forecasting accuracy is continuously improving on medium to large-scale watersheds, mainly through BD analysis and processing, by using ML models. However, there is currently no specialized theory and technology for forecasting floods in small watersheds, and the commonly used large and medium-sized watershed forecasting methods are not effective when applied to small watersheds, with insufficient advance forecasting. Generally, only temporary floods can be predicted, and the reliability of the forecasting cannot be measured, usually determined based on experience. The main reason is that there are many uncertain factors and short disaster time that cause small watershed floods, which leads to the lack of timeliness and reliability of AI forecasting. From a theoretical and technical perspective, firstly, the theory of forecasting floods in small watersheds tends to be model generalization, without considering the uncertainty of floods. Secondly, the connotation information of disaster causing factors is not fully utilized. The combination of RNNs and Bayesian improves the

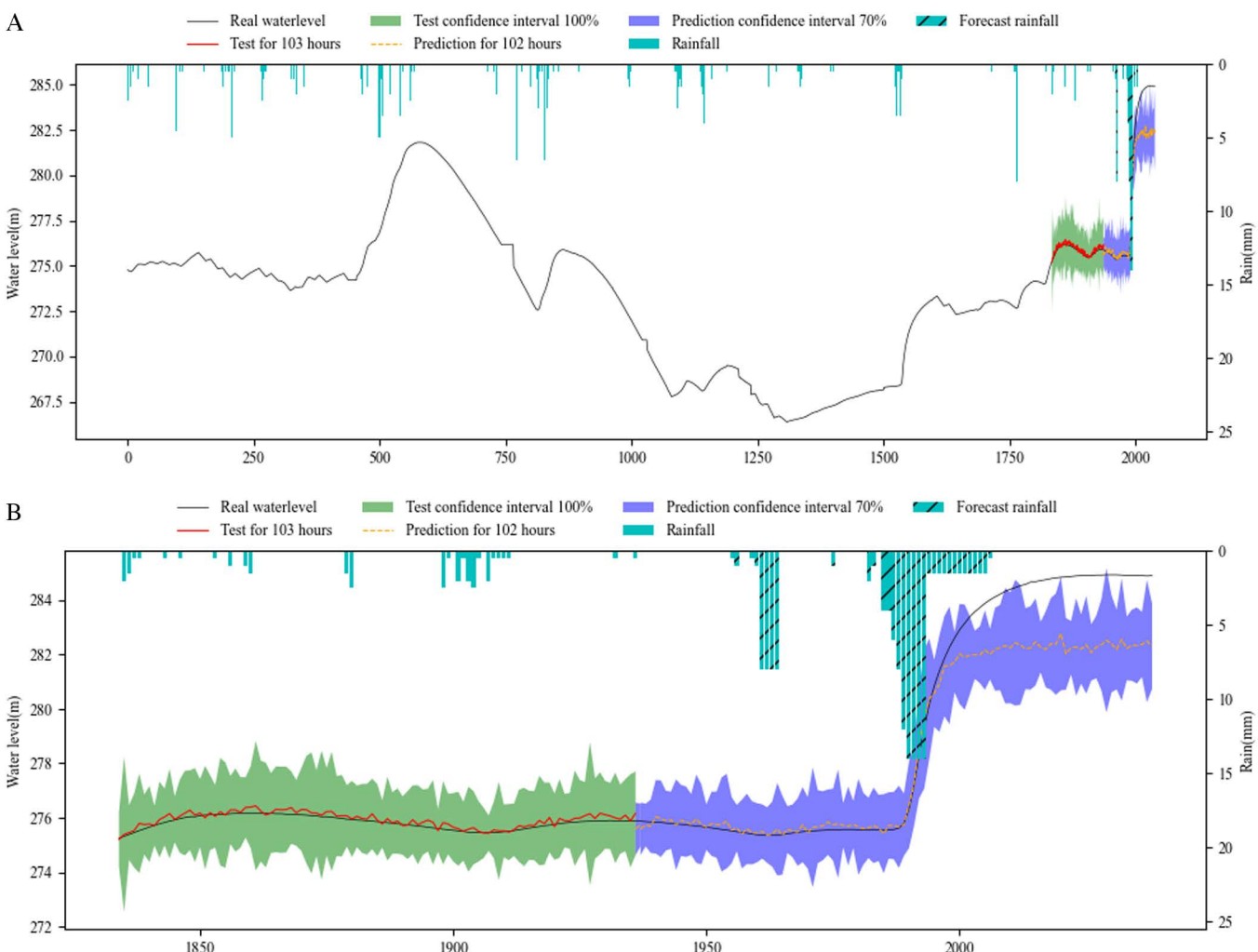

**Fig 7. Phase of LSTM-Bayesian model's test and prediction.** (a) Train, test and prediction phases' overall graph. (b) Test and prediction phases' detail graph.

reliability of flood water level prediction, and makes the scheduling of flood disasters reasonably evaluated.

Although the combination of the two fully reflects the reliability and accuracy, we need to consider the role of RNN series prediction methods, and even consider whether Bayesian reduces or improves the accuracy of prediction. We use the same data as the test part in this paper, and only use the RNN series part. The effect is shown in Fig 9.

Compared with the renderings of the experiment results in this paper, the accuracy of RNN alone is not as good as that combined with Bayesian. We calculated that Bayesian improved the accuracy of RNN series by about 15% on average, mainly because Bayesian method involves many RNN training to explore the optimal distribution of prediction probability and improve the accuracy of prediction. At the same time, it also shows that Bayesian method is effective. However, the training consumes a lot of time. Generally, the training of RNN series is less than 10 minutes, which is consistent with the literature [36]. However, our fastest GRU-Bayesian also takes 43 minutes.

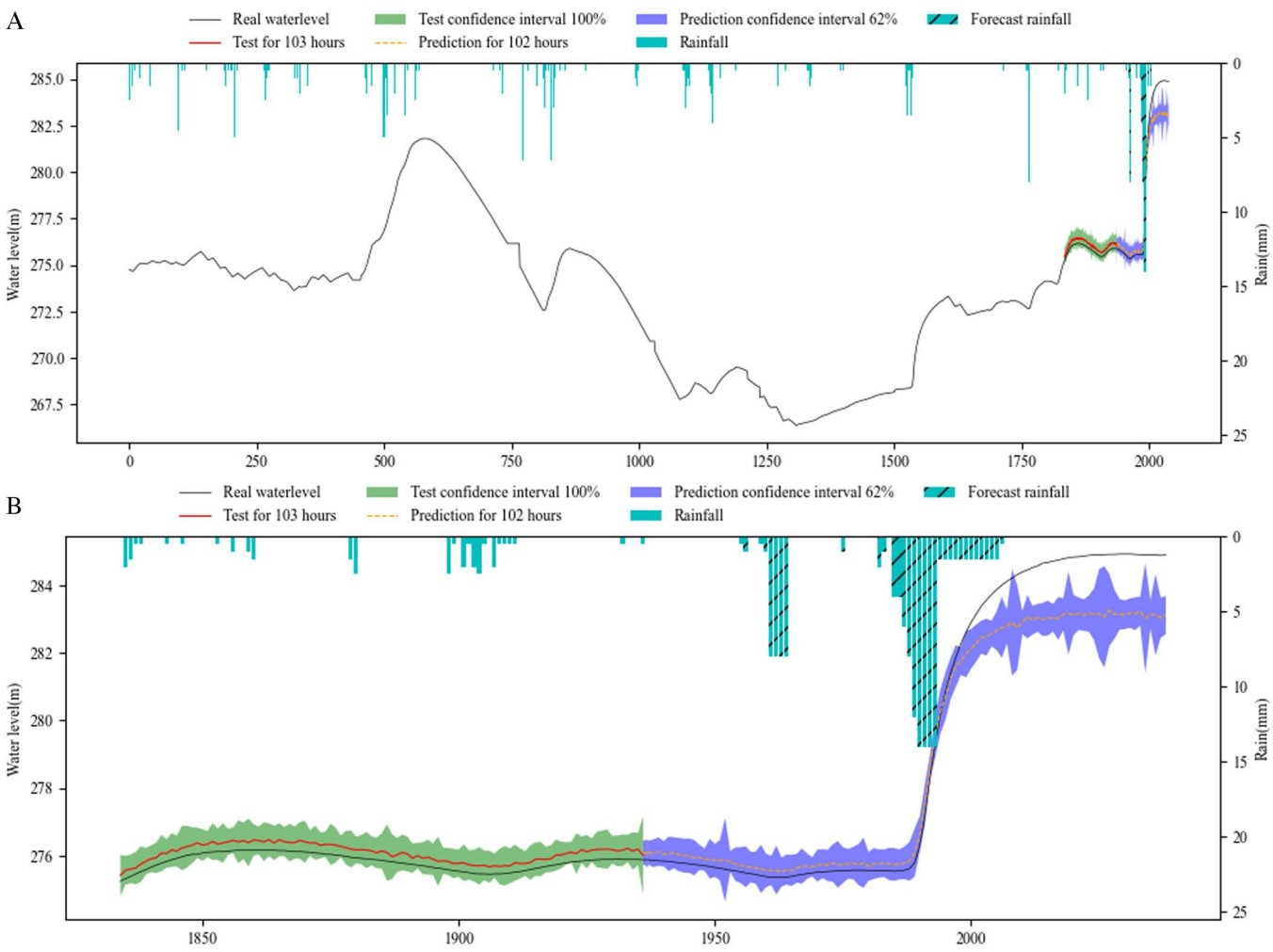

**Fig 8. Phases of GRU-Bayesian model's test and prediction.** (a) Train, test and prediction phases' overall graph. (b) Test and prediction phases' detail graph.

Table 6. Meaning of image identification about the results.

| Data | Meaning | Range | Position/Color |
|---|---|---|---|
| Water level | Water level Based on the elevation of National Vertical Datum 1985 of altitude datum | 260~290 | y-axis |
| Time series (Hours) | The data from March, 2023 to July, 2024 | 0~12000 | x-axis |
| Rainfall | Hourly rainfall (mm) | 0~25 | green |
| Real water level | The data for training of water level (m) | 260~290 | Black |
| Test for 103 hours | Confidence interval median for testing of water level (m) | 260~290 | Red |
| Test confidence interval | Confidence interval for prediction Within 103 hours (%) | 0~100% | Green |
| Forecast rainfall | The rain of predict by Meteorological department (mm) | 0~25 | Blue |
| Prediction confidence interval | Confidence interval for prediction Within 102 hours (%) | 0~100% | purple |
| Prediction for 102 hours | Confidence interval median for prediction of water level (m) | 260~290 | Orange |

**Table 7. Comparison of effects of RNNs-Bayesian's PICP, CIAW and MDAD at the phase of test and prediction.**

| Phase | Index | Base RNN-Bayesian | LSTM-Bayesian | GRU-Bayesian |
|---|---|---|---|---|
| Test | PICP | 69.03% | 100.00% | 100.00% |
| Test | CIAW | 1.82m | 2.08m | 0.68m |
| Test | MDAD | 1.11m | 0.01m | 0.09m |
| Prediction | PICP | All: 86.10%<br>0-72h: 86.32%<br>72-102h: 85.67% | All: 70.24%<br>0-72h: 96.18%<br>72-102h: 60.42% | All: 62.31%<br>0-72h: 90.03%<br>72-102h: 61.43% |
| Prediction | CIAW | All: 1.88m<br>0-72h: 1.42m<br>72-102h: 3.24m | All: 2.12m<br>0-72h: 1.83m<br>72-102h: 2.23m | All: 0.68m<br>0-72h: 0.43m<br>72-102h: 1.24m |
| Prediction | MDAD | All: 1.22m<br>0-72h: 1.13m<br>72-102h: 1.87m | All: 0.03m<br>0-72h: 0.01m<br>72-102h: 0.05.m | All: 0.13m<br>0-72h: 0.10m<br>72-102h: 0.17m |
| All | Reliability | 84.34% | 92.31% | 60.12% |
| All | Accuracy | 80.91% | 89.15% | 93.05% |

At the same time, the deterministic prediction RNNs can only get the definite value, cannot display the confidence interval, and the reliability of the prediction cannot be measured. Bayesian has the advantage in this respect. It is worth noting that since the Bayesian method lacks continuous prediction on the sequence between, the RNNs part of the algorithm fills this deficiency. It should be said that RNN and Bayesian interact and are indispensable.

Let's discuss whether only RNNs can play the prediction function of Bayesian. We use Random Forest Regression (RFR) and Bayesian combination. The effect is shown in Fig 10. RFR is an improvement over bagged decision tree. The decision tress that handles continuous variables are known as regression trees. Regression trees work similarly as decision trees by splitting each node based on a node impurity measure. We use the Sklearn Library (https://scikit-learn.org/), call the Application Programming Interface (API) of ML model directly. Through automatic parameter tuning, we obtained parameter (random_state = 60).

RFR belongs the regressive prediction, the circular part of RNNs cannot be used, and the information on time series cannot be obtained. The deterministic prediction part of RNNs is weak, and its effect becomes worse after superposition of Bayesian. For the current general-purpose attention-based models, we have tested the Transformer combined with Bayesian, as shown in Fig 11, and have not achieved good results. Of course, this does not mean that the transformer model is not suitable for prediction, whether short-term or long-term forecasting. The main reason is that it requires a lot of calculation and parameter adjustment. At present, it is less applied to time series prediction. The regression method of non RNN model cannot replace the deterministic prediction method of RNN series, which is the reason why RNNs are widely used in time series forecasting.

## 5.2 The time windows for forecasting

Rainfall is a primary determinant of water levels, particularly in extensive watersheds, due to the pronounced effects of rainfall convergence. In such contexts, the inaccuracies associated with seepage are unlikely to significantly influence the ultimate water level. Conversely, in small watersheds, the relationship between rapid rainfall rates and geographical factors, wind direction, and anthropogenic structures is characterized by nonlinearity and uncertainty. While rainfall remains a critical variable, it is essential to acknowledge the importance of other contributing factors.

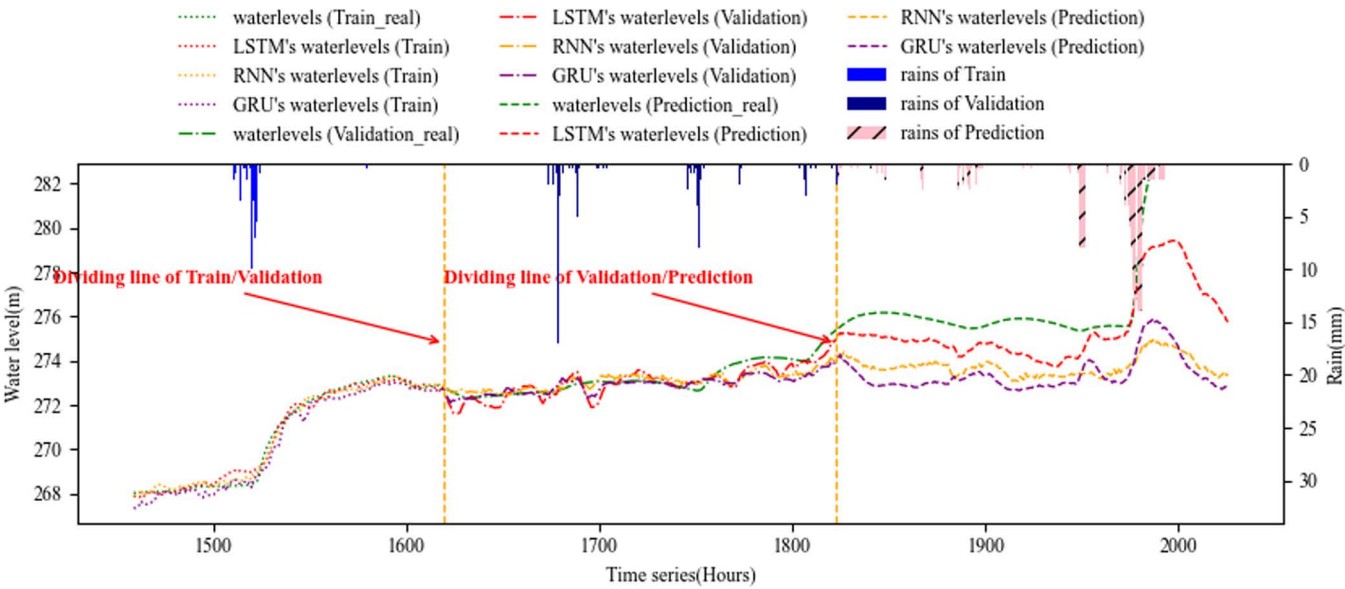

**Fig 9. Phases of RNN models' train, validation and prediction.**

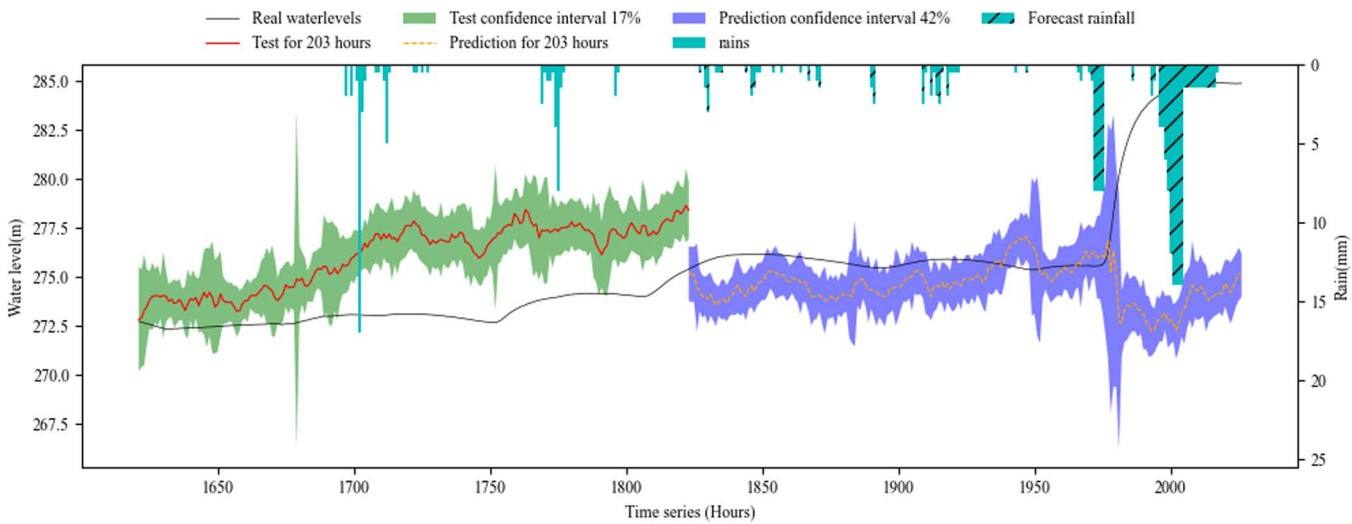

**Fig 10. Phases of RFR-Bayesian model's test and prediction.**

Moreover, an excessive focus on historical disaster-inducing factors—such as those from several weeks or even months prior—can lead to substantial changes in geographical elements (including terrain and groundwater content) and seasonal wind patterns. Consequently, the relevance of historical data diminishes significantly. To enhance the accuracy of disaster water level predictions, it is imperative to utilize data pertaining to disaster-inducing factors from recent days. Our experimental comparisons indicate that an increase in the volume of training data does not necessarily correlate with improved prediction accuracy. As illustrated in Table 8, the MDAD of various models is minimized when the time window is set to five days. Both

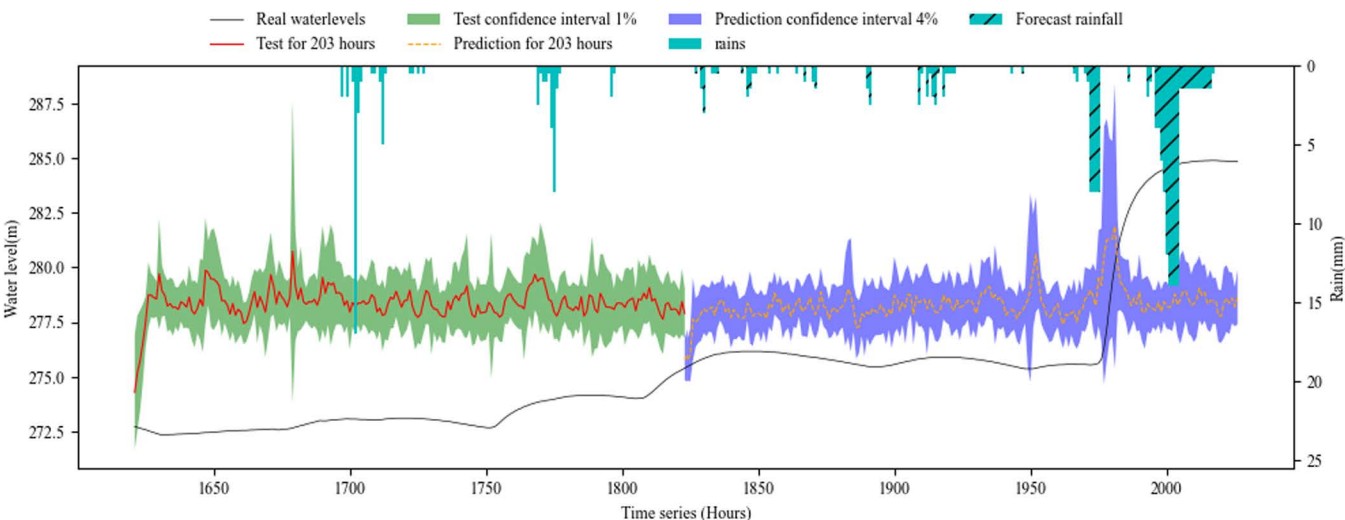

**Fig 11. Phases of Transformer-Bayesian model's test and prediction.**

excessively short and excessively long time windows are detrimental to time series prediction accuracy.

The RNNs model employed in this study effectively leverages recent data to enhance predictive accuracy. Furthermore, this approach is applicable in scenarios with extensive prediction instances, thereby augmenting its adaptability.

### 5.3 The coordination between data and models

Small watershed flood is mainly influenced by natural factors. From the perspective of the feasibility of intelligent technology and the continuous improvement of forecasting accuracy, the use of BD and AI methods has certain advantages. RNNs are currently a highly advanced AI time series forecasting technology, with the most prominent being the multi-dimensional and multi-step RNNs time series forecasting technology by using RNNs and Bayesian probability. In the field of water conservancy forecasting, another issue involves complex data sources and high dimensions, making it difficult to identify effective information that causes disasters. However, adopting the latest water conservancy BD effective information extraction technology can further enhance the reliability of RNNs.

The disaster causing factors are very important. The more detailed and accurate the data, the more reliable and accurate the forecasting effect of the AI method will be. The traditional Auto Regressive Integrate Moving Average Model (ARIMA) [38] model does not pay attention to the influence factors, but only based on the change law of the forecasting itself, which not only has large errors, but also has poor interpretability. Without the model based on the influence factors, it is not convincing. Its essence is that the factors-based forecasting needs the support of effective BD. However, the data for flood forecasting come from many kinds of real-time BD, such as hydrology, meteorology, geography, crowdsourcing and so on, and are related to many monitoring points in the upstream of each tributary. The real-time acquisition of multimodal BD has not been carried out on a large scale in the water conservancy industry, which is also the reason for the generalized calculation of traditional physical models.

The data-driven models are difficult to measure the interpretability. We call it in line with the physical law of water level change, that is, the degree of influence by disaster causing

factors. The hydrological, meteorological, geographical and crowdsourcing data are considered in this paper. We explain from data, model, effect and sustainability. As shown in Table 9, the H, M and L represent High, Middle and Low. Breaking through the research on the critical rainfall and physical models caused by floods, this is the first time that the basic theory and key technologies of basin flood disaster forecasting based on RNNs have been studied from the perspective of uncertainty forecasting, improving the interpretability and timely reliability of flood disaster forecasting. Research on the methods of multi-dimensional and BD driven and composite AI modeling to solve complex problems. By integrating multi-dimensional data of flood disaster factors, effective disaster time series information is extracted, and an RNNs forecasting model that integrates certainty and uncertainty forecasting is established to improve the reliability of flood disaster forecasting.

The advantage of RNNs-Bayesian is that they are inclusive of the accuracy of data. This is also the advantage of combining BD as follows.

(1) We have constructed a forecasting mechanism for the transfer of spatiotemporal factors caused by small watershed floods to disaster uncertainty, extracted the theoretical impact relationship between disaster factors and disaster results, and established a BD & RNNs-Bayesian forecasting theoretical method.

(2) By real-time evaluation of multi-source and multi-dimensional disaster causing factors, adopting feature dynamic selection and time-frequency fusion, the effectiveness of disaster information has been improved, providing effective disaster information for predicting floods in river basins with missing data and low forecasting density.

## 5.4 Reliability and accuracy

Bayesian methods are associated with reliability, while RNNs are linked to accuracy. The integration of these two approaches facilitates reliable and precise data forecasting. However, these objectives are inherently contradictory; enhancing reliability may lead to a reduction in accuracy, and vice versa. Achieving an optimal balance between these two goals necessitates their consideration during model training to maximize overall performance. The use of LSTM has been shown to significantly enhance accuracy and reliability.

As the prediction steps increases, according to Figs 6,7 and 8, the prediction confidence interval is getting larger and larger, but the ability to cover the real value is getting lower and lower, both reliability and accuracy tend to decline, a phenomenon that aligns with established principles regarding uncertainty. The reliability is not guaranteed, and the accuracy is getting lower and lower. In addition, according to Table 8, when the prediction window is the 5 days, the MDAD is the smallest. The bigger it is, the worse the comprehensive performance of reliability and accuracy is.

There exists a need to address the limitations of the Bayesian algorithm to bolster forecasting reliability. Typically, as the forecasting horizon extends, the accumulation of errors in base RNN-Bayesian models results in deteriorating accuracy. In contrast, LSTM-Bayesian models

**Table 8. The relationship between MDAD of prediction and the length of time window.**

| Number of days | 1 | 3 | 5 | 7 | 10 |
|---|---|---|---|---|---|
| Base RNN-Bayesian | 2.02 | 1.71 | 1.22 | 1.67 | 1.91 |
| LSTM-Bayesian | 0.1 | 0.05 | 0.03 | 0.05 | 0.08 |
| GRU-Bayesian | 0.81 | 0.24 | 0.13 | 0.15 | 0.21 |
| RFR-Bayesian | 5.13 | 3.15 | 2.24 | 2.56 | 2.71 |

**Table 9. Comparison of effects of RNNs-Bayesian model.**

| Model | Data | | | Model | | Basic effect | | | Sustainability | | | Ref |
|---|---|---|---|---|---|---|---|---|---|---|---|---|
| | Data set | Characteristics | Prepro-cessing | Charac-teristics | Results | Time-liness | Reli-abil-ity | Accu-racy | Error trend | Operability in Small Watersheds | Interpret-ability | |
| Rainfall threshold | Rainfall | Small amount of data, single dimension | / | Accord-ing to the threshold | Will be (or not) | L | M | L | The error increases gradually | H | simple | [1,9] |
| Hydrology | Rainfall, underlying surface characteris-tics, runoff, experience, geographic informa-tion, etc. | Small amount of data, multi-dimensional | / | Physical model for simulating | Continu-ous value of water level or flow | L | H | M | The factors of small water-shed change rapidly, and the calcu-lation error increases | L, Runoff and other data are not easy to measure | Can simu-late physical phenomena | [17,18,20] |
| Statistics | Water level over the years | Large amount of data, single dimension | Large amount of data, data cleaning | Fixed statistical model | Continu-ous value of water level or flow | M | L | H | The factors of small water-shed change rapidly, and the data regu-larity has low reproducibil-ity and large error | H | Statistical interpreta-tion only | [1,5,8] |
| Our method: RNNs-Bayesian | Hydrology, meteo-rology, geography, human activities and other multidi-mensional data | Large amount of data, multi-dimensional | Data cleaning, feature analy-sis and extraction | Multidi-mensional dynamic probabil-ity model, constantly updated | Continu-ous value of water level and flow, Confi-dence interval value | H | H | H | Rolling forecast-ing model, which is continuously updated, maintains the accuracy of forecasting with the com-prehensive conclusion of certainty and uncertainty forecast-ing, and has strong sustainability | H, Available rainfall and water level stations and meteo-rological forecasting information | Compre-hensive effect of uncertain multidi-mensional impact factors, and the con-clusion of confidence interval is more scientific than the conclusion of certainty | [19,36] |

have demonstrated favorable outcomes, primarily due to LSTM's ability to extract information through feedback loops, which mitigates errors and enhances forecasting precision. Further-more, the computational complexity and time associated with the entire dynamic process are manageable, allowing for concurrent application across numerous small watershed water level forecasting points, thereby yielding cross-forecasting and more accurate results.

Experimental analyses indicate that while the overall confidence interval is maintained, accurate prediction of water level transient, but the peak forecasting performance remains suboptimal, which is a critical metric for water level forecasting. To illustrate the complexities associated with fluctuations in flood water levels, it is noteworthy that existing literature [31] often relies on data collection periods that are either annual or monthly, characterized by regularity, low perplexity, and relatively low forecasting difficulty. Consequently, such datasets are ill-suited for small watershed flood water level with irregular changes forecasting.

It is important to acknowledge that the RNNs-Bayesian models proposed in this study is not without its flaws. To sustain model accuracy, data preprocessing is essential, and continuous training and calibration of the model are required, failure to do so may result in significant error accumulation, ultimately compromising both reliability and accuracy in forecasting.

Moreover, in scenarios involving complex water level fluctuations, numerous studies reveal relatively periodic data changes on a large scale, such as yearly, monthly, weekly, and daily intervals. Although similarities may exist across years, actual water level patterns exhibit low periodicity, reflecting real-world conditions. This type of forecasting presents a considerable degree of complexity and is often more challenging than forecasting stock data, occasionally approaching the level of noise. Additionally, as the forecasting time frame extends, the accumulation of errors exacerbates, leading to a progressive decline in both reliability and accuracy.

The analysis presented in the referenced study [5] indicates that the omission of soil moisture data can result in significant inaccuracies in flood volume estimations. Among the various meteorological parameters, rainfall and duration are identified as critical factors for predicting sewer flooding. While the inclusion of additional data may enhance the reliability and accuracy of flood forecasts, it is essential to recognize the inherent tension between these two attributes. Enhancing the reliability of flood forecasting may inadvertently compromise accuracy, thereby undermining the primary objective of effective forecasting. Conversely, if efforts to improve accuracy lead to a decline in reliability, such approaches are unlikely to be embraced within the water conservancy sector, where the primary focus is on safeguarding public safety through disaster prevention and mitigation strategies.

Furthermore, a notable limitation of this study is that, despite the reliance on hydrological models—which offer greater interpretability compared to conventional data-driven models—there remains a need for further enhancement of interpretability. Achieving a level of interpretability comparable to that of physical estimation models is crucial for advancing hydrological forecasting methodologies.

## 6 Conclusions

We investigated the method of the RNNs-Bayesian model in flood forecasting. Due to the multiple uncertain factors caused by flood, rapid changes in spatiotemporal factors, and rapid disaster occurrence, the reliability of deterministic flood forecasting is insufficient. We study the forecasting mechanism of the transfer of spatiotemporal factors caused by multiple sources of floods to disaster uncertainty, use the BD and confidence interval forecasting model, a fast and high-performance spatiotemporal uncertainty forecasting method is constructed. Confidence intervals are used to achieve timely and reliable forecasting of floods. The main conclusions are as follows.

(1) A notable advancement of this study is the establishment of a methodology that successfully integrates time series accuracy forecasting with probabilistic evaluations in the realm of small watershed flood forecasting. Through the combination of RNNs and Bayesian principles, we have formulated a confidence interval forecasting model that improves the reliability and precision of flood predictions. Especially, the effect of LSTM-Bayesian is the best.

(2) The main evaluation indicator for confidence interval forecasting is the average value, but currently there is less attention paid to the peak value, while the focus of water level forecasting is often on the peak value. We have made improvements in this regard. At the same time, it improves the real-time and interpretability.

(3) Accuracy and reliability are a pair of contradictions that need to be considered simultaneously. The coverage and width of the confidence interval can be adjusted to improve reliability and accuracy.

The biggest contribution of the confidence interval forecasting model of small watered flood based on compound RNN and Bayesian proposed in this paper is that the RNNs-Bayesian models can consider both reliability and accuracy of flood water level prediction, which is also of great concern in hydrological forecasting. Of course, it should also be considered that the model relies on the high-quality near time series data of the monitoring points' prediction time points, and has excellent prediction effect for the short term (0~72h), but the reliability and accuracy of the medium and long-term prediction are not enough. The uncertain prediction model will play an important role in the prediction, early warning, preview and pre planning of small watershed. Furthermore, through large-scale applications and intelligent training, continuous performance improvement can be achieved, analyze and organize the forecasting elements based on the Large Language Model (LLM) to improve the application performance of the data, and improving the interpretability and timely reliability of long-term forecasting models.

## Author contributions

**Data curation:** Songsong Wang.

**Funding acquisition:** Ouguan XU.

**Methodology:** Songsong Wang.

**Resources:** Ouguan XU.

**Validation:** Songsong Wang.

**Visualization:** Songsong Wang.

**Writing – original draft:** Songsong Wang.

**Writing – review & editing:** Songsong Wang.

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
