## [Decision Letter · Decision Letter 0]

29 Nov 2024

PONE-D-24-42128Confidence interval forecasting model of small watershed flood based on compound Recurrent Neural Networks and BayesianPLOS ONE

Dear Dr. Wang,

Thank you for submitting your manuscript to PLOS ONE. After careful consideration, we feel that it has merit but does not fully meet PLOS ONE’s publication criteria as it currently stands. Therefore, we invite you to submit a revised version of the manuscript that addresses the points raised during the review process.

We look forward to receiving your revised manuscript.

Kind regards,

Jinran Wu, PhD

Academic Editor

PLOS ONE

“This work was supported by Joint Funds of the Zhejiang Provincial Natural Science Foundation of China (Grant No. LZJWZ23E090001), and Ministry of Education Humanities and Social Sciences of China (Grant No. 22YJCZH162).”

“This work was supported by Joint Funds of the Zhejiang Provincial Natural Science Foundation of China (Grant No. LZJWZ23E090001), and Ministry of Education Humanities and Social Sciences of China (Grant No. 22YJCZH162).”

“This work was supported by Joint Funds of the Zhejiang Provincial Natural Science Foundation of China (Grant No. LZJWZ23E090001), and Ministry of Education Humanities and Social Sciences of China (Grant No. 22YJCZH162).”

Reviewers' comments:

Reviewer's Responses to Questions

**Comments to the Author**

1. Is the manuscript technically sound, and do the data support the conclusions?

Reviewer #1: Yes

2. Has the statistical analysis been performed appropriately and rigorously? 

Reviewer #1: Yes

3. Have the authors made all data underlying the findings in their manuscript fully available?

Reviewer #1: Yes

4. Is the manuscript presented in an intelligible fashion and written in standard English?

Reviewer #1: Yes

5. Review Comments to the Author

Reviewer #1: Dear Editor,

Thank you for the opportunity to review this paper. I believe the authors' approach of combining RNN with Bayesian estimation for quantile regression is quite innovative, and their argumentation is clear. However, I have a few suggestions for consideration:

1. Lack of comprehensive comparison experiments: The authors only compared the performance of three different models within the RNN+Bayesian framework. Methods such as ensemble approaches and attention-based models are also commonly used in time-series forecasting tasks. Including these comparison experiments would more strongly demonstrate the practical value of the proposed model.

2. Lack of ablation experiments: Ablation experiments would make the authors' arguments more well-rounded. For instance, comparing RNN+LSTM with using only LSTM could help illustrate the difference and thereby justify the use of Bayesian estimation.

3. Some parts lack clarity: It would be best for the authors to clearly and prominently list all contributions of this paper in one section. Additionally, it should be explicitly stated that the model in the paper predicts n steps ahead based on the previous m steps. Furthermore, the authors could enhance the paper by discussing the model's performance on data with varying sequence lengths.

4. The authors should clarify the meaning of the x-axis labels in the figures.

Overall, this is a valuable paper. If the authors can address and improve upon the issues mentioned above, I recommend it for publication.

6. PLOS authors have the option to publish the peer review history of their article (what does this mean? ). If published, this will include your full peer review and any attached files.

**Do you want your identity to be public for this peer review?** For information about this choice, including consent withdrawal, please see our Privacy Policy .

Reviewer #1: No

---

## [Author Response · Author response to Decision Letter 1]

31 Dec 2024

Original Article Title: “Confidence interval forecasting model of small watershed flood based on compound Recurrent Neural Networks and Bayesian” (PONE-D-24-42128)

To: PLOS ONE Editor

Re: Response to reviewers

Dear reviewers and editors,

Thank you for allowing a resubmission of our manuscript, with an opportunity to address the reviewers’ comments.

We are uploading (a) our point-by-point response to the comments (below) (response to reviewers), (b) an updated manuscript with red highlighting indicating changes (Supplementary Material for Review), and (c) a clean updated manuscript without highlights (Main Manuscript).

Best regards,

Songsong Wang, Ouguan Xu

Reviewer#1, Concern # 1: Lack of comprehensive comparison experiments: The authors only compared the performance of three different models within the RNN+Bayesian framework. Methods such as ensemble approaches and attention-based models are also commonly used in time-series forecasting tasks. Including these comparison experiments would more strongly demonstrate the practical value of the proposed model.

Author response: Thank you for pointing this out. The reviewer is correct, and we have revised it.

Author action: We updated the relevant contents in the paper as follows.

We have employed various integrated regression of Machine Learning (ML) in conjunction with Bayesian methods to ascertain the optimal combination for forecasting.

We use Random Forest Regression (RFR) and Bayesian combination. RFR is an improvement over bagged decision tree. The decision tress that handles continuous variables are known as regression trees. Regression trees work similarly as decision trees by splitting each node based on a node impurity measure. We use the Sklearn Library (https://scikit-learn.org/), call the Application Programming Interface (API) of ML model directly. Through automatic parameter tuning, we obtained parameter (random_state = 60). RFR belongs the regressive prediction, the circular part of RNNs cannot be used, and the information on time series cannot be obtained. The deterministic prediction part of RNNs is weak, and its effect becomes worse after superposition of Bayesian.

For the current general-purpose attention-based models, we have tested the Transformer combined with Bayesian, and have not achieved good results. Of course, this does not mean that the transformer model is not suitable for prediction. The main reason is that it requires a lot of calculation and parameter adjustment. At present, it is less applied to time series prediction. The regression method of non RNN model cannot replace the deterministic prediction method of RNN series, which is the reason why RNN series is widely used in time series forecasting.

Reviewer#1, Concern # 2: Lack of ablation experiments: Ablation experiments would make the authors' arguments more well-rounded. For instance, comparing RNN+LSTM with using only LSTM could help illustrate the difference and thereby justify the use of Bayesian estimation.

Author response: Thank you for pointing this out. The reviewer is correct, and we have revised it.

Author action: We updated the relevant contents in the paper as follows.

The combination of RNNs and Bayesian improves the reliability of flood water level prediction, and makes the scheduling of flood disasters reasonably evaluated.

Although the combination of the two fully reflects the reliability and accuracy, we need to consider the role of RNN series prediction methods, and even consider whether Bayesian reduces or improves the accuracy of prediction. We use the same data as the test part in this paper, and only use the RNN series part.

Compared with the renderings of the experiment results in this paper, the accuracy of RNN alone is not as good as that combined with Bayesian. We calculated that Bayesian improved the accuracy of RNN series by about 15% on average, mainly because Bayesian method involves many RNN training to explore the optimal distribution of prediction probability and improve the accuracy of prediction. At the same time, it also shows that Bayesian method is effective. However, the training consumes a lot of time. Generally, the training of RNN series is less than 10 minutes, which is consistent with the literature [37]. However, our fastest GRU-Bayesian also takes 43 minutes.

At the same time, the deterministic prediction RNNs can only get the definite value, cannot display the confidence interval, and the reliability of the prediction cannot be measured. Bayesian has the advantage in this respect. It is worth noting that since the Bayesian method lacks continuous prediction on the sequence between, the RNNs part of the algorithm fills this deficiency. It should be said that RNN and Bayesian interact and are indispensable.

Reviewer#1, Concern # 3: Some parts lack clarity: It would be best for the authors to clearly and prominently list all contributions of this paper in one section. Additionally, it should be explicitly stated that the model in the paper predicts n steps ahead based on the previous m steps. Furthermore, the authors could enhance the paper by discussing the model's performance on data with varying sequence lengths.

Author response: Thank you for pointing this out. The reviewer is correct, and we have revised it.

Author action: We updated the relevant contents in the part of Introduction.

Through an examination of the uncertainty theory pertaining to forecasting and testing, it has been determined that the forecasting capabilities of the proposed model remain within the bounds of the reliability forecasting interval, even during periods characterized by rapid fluctuations in water levels. This model achieves a commendable equilibrium between reliability and accuracy. The approach employed involves the utilization of a multidimensional disaster data input unit for water level forecasting, with LSTM networks identified as the most effective method for confidence interval forecasting. Additionally, we have employed various integrated regression of Machine Learning (ML) in conjunction with Bayesian methods to ascertain the optimal combination for forecasting. A significant contribution of this research lies in the identification of a methodology that effectively merges time series accuracy forecasting with probabilistic assessments within the context of small watershed flood forecasting. By integrating RNNs with Bayesian principles, we have developed a confidence interval forecasting model that enhances both the reliability and accuracy of flood forecasting.

We use the Time window predicts next step on the previous 120 steps. About time window, the data of flood water level at the future time point (each time is the next hour) is predicted through the data of disaster causing factors in the moving time window.

Our experimental comparisons indicate that an increase in the volume of training data does not necessarily correlate with improved prediction accuracy. As illustrated in Table 8, the MDAD of various models is minimized when the time window is set to five days. Both excessively short and excessively long time windows are detrimental to time series prediction accuracy.

Table 8 The relationship between MDAD of prediction and the length of time window

Number of days 1 3 5 7 10

Base RNN-Bayesian 2.02 1.71 1.22 1.67 1.91

LSTM-Bayesian 0.1 0.05 0.03 0.05 0.08

GRU-Bayesian 0.81 0.24 0.13 0.15 0.21

RFR-Bayesian 5.13 3.15 2.24 2.56 2.71

Reviewer#1, Concern # 4: The authors should clarify the meaning of the x-axis labels in the figures.

Author response: Thank you for pointing this out. The reviewer is correct, and we have revised it.

Author action: We updated the relevant figures in the paper.

---

## [Decision Letter · Decision Letter 1]

10 Jan 2025

PONE-D-24-42128R1Confidence interval forecasting model of small watershed flood based on compound Recurrent Neural Networks and BayesianPLOS ONE

Dear Dr. Wang,

Thank you for submitting your manuscript to PLOS ONE. After careful consideration, we feel that it has merit but does not fully meet PLOS ONE’s publication criteria as it currently stands. Therefore, we invite you to submit a revised version of the manuscript that addresses the points raised during the review process.

We look forward to receiving your revised manuscript.

Kind regards,

Jinran Wu, PhD

Academic Editor

PLOS ONE

Journal Requirements:

Reviewers' comments:

Reviewer's Responses to Questions

**Comments to the Author**

1. If the authors have adequately addressed your comments raised in a previous round of review and you feel that this manuscript is now acceptable for publication, you may indicate that here to bypass the “Comments to the Author” section, enter your conflict of interest statement in the “Confidential to Editor” section, and submit your "Accept" recommendation.

Reviewer #2: (No Response)

Reviewer #3: (No Response)

2. Is the manuscript technically sound, and do the data support the conclusions?

Reviewer #2: (No Response)

Reviewer #3: (No Response)

3. Has the statistical analysis been performed appropriately and rigorously? 

Reviewer #2: (No Response)

Reviewer #3: (No Response)

4. Have the authors made all data underlying the findings in their manuscript fully available?

Reviewer #2: (No Response)

Reviewer #3: (No Response)

5. Is the manuscript presented in an intelligible fashion and written in standard English?

Reviewer #2: (No Response)

Reviewer #3: (No Response)

6. Review Comments to the Author

Reviewer #2: Reviewer Comments on Manuscript Number: PONE-D-24-42128R1

This research aims to improve the accuracy and reliability of flood and water level forecasting in small watersheds. The methods used in this study are novel and precise, making this research valuable. However, the practical aspects of the model have not been adequately addressed. My detailed comments on this article are as follows:

1. Abstract: The abstract seems insufficient and lacks clarity. It does not explain the significance and practical application of the models, the input variables, the length of the statistical period, and the time scale. Adding one or two sentences could resolve these ambiguities.

2. Structure: Unfortunately, the article does not follow the IMRAD (Introduction, Methods, Results, and Discussion) structure. Overall sectioning and subsections would help clarify the topic and improve understandability.

3. Study Area Map: A map of the study area (also DEM, Land use/Land Cover maps) is not provided, nor is there sufficient explanation regarding the characteristics of the study area.

4. Statistical Characteristics: It would have been better to specify the statistical characteristics of the data used in a table.

5. Variable Combinations: The reasons for selecting combinations of input model variables should be explained, and the scenarios used should be specified in a table, including the combinations of input parameters.

6. Correlation Matrix: The correlation matrix between input and output variables is not provided.

7. Evaluation Criteria: The evaluation criteria for the models used have not been presented. It would have been better to provide these criteria values for each model in a table to compare the results.

8. Conclusion Section: In the general conclusion section, it would have been better to mention the strengths and weaknesses of the model, as well as its advantages and disadvantages.

9. Overall Suitability: In summary, I find the article suitable for publication after making these revisions.

Reviewer #3: The paper has proposed Confidence interval forecasting model of small watershed flood using compound Recurrent Neural Networks and Bayesian. These are several issues associated with the presented work:

1. Write the abstract more clear with some output values.

2. Contribution is not clear.

3. In the introduction and conclusion, emphasize the practical applications and real world impact of the proposed model to align the technical work with practical significance.

4. Analyse the performance of the algorithm as the prediction steps increases.

For above the following articles can be addressed:

10.17775/CSEEJPES.2016.00970

https://doi.org/10.1007/s12145-023-01020-9

---

## [Author Response · Author response to Decision Letter 2]

6 Mar 2025

Reviewer#2, Concern # 1: Abstract: The abstract seems insufficient and lacks clarity. It does not explain the significance and practical application of the models, the input variables, the length of the statistical period, and the time scale. Adding one or two sentences could resolve these ambiguities.

Author response: Thank you for pointing this out. The reviewer is correct, and we have revised it.

Author action: We updated the relevant contents in the paper as follows.

We used the multidimensional disaster data input unit for water level forecasting, including hydrology, meteorology, and geography, and 5 days of time windows for forecasting, The comprehensive reliability of LSTM-Bayesian for 0~102 hours flood reached 92.31%, and the comprehensive accuracy reached 89.15%, and confidence interval forecasting using LSTM is the best method, and achieved reasonable balance of reliability and accuracy. Overall, compound RNN could be a good alternative for forecasting hourly streamflow and extreme water level in small watersheds.

Reviewer#2, Concern # 2: Structure: Unfortunately, the article does not follow the IMRAD (Introduction, Methods, Results, and Discussion) structure. Overall sectioning and subsections would help clarify the topic and improve understandability.

Author response: Thank you for pointing this out. The reviewer is correct, and we have revised it.

Author action: We updated the relevant contents in the paper as follows.

(1) We have revised the "Methods" section, "3.2 RNNs Bayesian model", to make it clearer.

(2) (2) We merged data and results to generate the "4 Data and results" chapter.

(3) Overall sectioning and subsections would help clarify the topic and improve understandability.

Reviewer#2, Concern # 3: Study Area Map: A map of the study area (also DEM, Land use/Land Cover maps) is not provided, nor is there sufficient explanation regarding the characteristics of the study area.

Author response: Thank you for pointing this out. The reviewer is correct, and we have revised it.

Author action: We updated the relevant contents in the paper as follows.

(1) We acknowledge that creating high-quality maps does help enhance the visualization of articles. However, in the actual operation process, we have encountered some technical and resource challenges: lack of professional mapping tools and technical capabilities, team members do not have the operational skills of advanced GIS software, and it is difficult to produce maps that meet the publishing standards in the short term. Considering the time arrangement and budget constraints of the project, it is difficult to obtain and process the required high-resolution remote sensing images and terrain data. In order to make up for the lack of providing maps directly, we added geographical introduction to help readers better understand the research background and results.

(2) We added a full explanation of the characteristics of the study area：The upper reaches of the Qiantang River are characterized by mountainous terrain, with the Qixi Reservoir monitoring point located in Kaihua County, Zhejiang Province, China, which marks the source of the Qiantang River. Kaihua County experiences a subtropical monsoon climate, situated on the northern edge of this climatic zone. This region is known for its warmth, humidity, abundant rainfall, and distinct four seasons. The annual average precipitation in Kaihua County is 1990 mm, ranking second highest in Zhejiang Province.

Rainfall in the Qiantang River watershed is primarily concentrated during the summer months (June to September), particularly during the plum rain and typhoon seasons. Heavy rainfall during these periods is the primary cause of floods in the region. The upper reaches of the Qiantang River feature steep terrain, narrow river valleys, and rapid water flow, conditions that facilitate the rapid formation of floods. Floods often develop within a short time frame, typically hours or even minutes after heavy rainfall, posing significant challenges for early warning systems and emergency response efforts.

Reviewer#2, Concern # 4: Statistical Characteristics: It would have been better to specify the statistical characteristics of the data used in a table.

Author response: Thank you for pointing this out. The reviewer is correct, and we have revised it.

Author action: We updated the relevant contents in the paper as follows.

We added the accuracy and reliability features, and explained and discussed them.

Reviewer#2, Concern # 5: Variable Combinations: The reasons for selecting combinations of input model variables should be explained, and the scenarios used should be specified in a table, including the combinations of input parameters.

Author response: Thank you for pointing this out. The reviewer is correct, and we have revised it.

Author action: We updated the relevant contents in the paper as follows. We detailed description of the problem.

Reviewer#2, Concern # 6: Correlation Matrix: The correlation matrix between input and output variables is not provided.

Author response: Thank you for pointing this out. The reviewer is correct, and we have revised it.

Author action: We updated the relevant contents in the paper as follows.

There are many disaster-causing factors affecting future water level prediction, including hydrology, meteorology and geography, etc. In order to obtain the factors with great influence, we use the relationship matrix, add the relationship matrix to the analysis of the relationship between the flood disaster-causing factors and the water level, as shown in Figure 4.

The future water level to be predicted is mainly related to the historical water level, up to 58%, and the relationship with rainfall is up to 28%, which is consistent with the basic law of hydrology. In addition, humidity, weather status, geography, etc. are also closely related to the future water level, and these data will be used as the input elements of the uncertainty prediction model.

Reviewer#2, Concern # 7: Evaluation Criteria: The evaluation criteria for the models used have not been presented. It would have been better to provide these criteria values for each model in a table to compare the results.

Author response: Thank you for pointing this out. The reviewer is correct, and we have revised it.

Author action: We updated the relevant contents in the paper as follows. e detailed description of the problem.

In the key phase of prediction, LSTM-Bayesian still maintained the advantages of the test phase, especially in the 0-72h phase, the PICP reached 96.18%, which was the closest to the actual water level, and the MDAD was only 0.01m, which had a good prediction effect, taking into account the reliability and accuracy. GRU-Bayesian also performed well, the only disadvantage is that MDAD still exists large error, and 10 times less than LSTM-Bayesian, but it is also within the acceptable range. The deviation of base RNN-Bayesian is the weakest among the three. Another obvious feature is that when the water level rises rapidly, the prediction effect is better, and the main disaster causing factors, mainly rainfall, are considered in the model. In the 72-102h phase, LSTM-Bayesian and GRU-Bayesian had a large deviation, and PICP could not cover the actual water level. base RNN-Bayesian covered the actual water level due to PICP with a large coverage rate, and the MDAD value was relatively stable. At the same time, the applicability of water level prediction was reduced due to too large PICP. In the prediction phase, 0-72h maintained the advantage, and the three predictions were effective.

To facilitate statistics and analysis, we convert PICP and CIAW into comprehensive reliability and accuracy. The reliability of base RNN-Bayesian and LSTM-Bayesian models is higher than GRU-Bayesian model, reaching 84.34% and 92.31%, respectively. The accuracy of LSTM Bayesian and GRU-Bayesian prediction models is higher than base RNN-Bayesian model, reaching 89.15% and 93.05%, respectively. Taken together, LSTM-Bayesian model has advantages in reliability and accuracy.

Reviewer#2, Concern # 8: Conclusion Section: In the general conclusion section, it would have been better to mention the strengths and weaknesses of the model, as well as its advantages and disadvantages.

Author response: Thank you for pointing this out. The reviewer is correct, and we have revised it.

Author action: We updated the relevant contents in the paper as follows.

The biggest contribution of the confidence interval forecasting model of small watered flood based on compound RNN and Bayesian proposed in this paper is that the RNNs-Bayesian models can consider both reliability and accuracy of flood water level prediction, which is also of great concern in hydrological forecasting. Of course, it should also be considered that the model relies on the high-quality near time series data of the monitoring points' prediction time points, and has excellent prediction effect for the short term (0~72h), but the reliability and accuracy of the medium and long-term prediction are not enough. The uncertain prediction model will play an important role in the prediction, early warning, preview and pre planning of small watershed.

Reviewer#2, Concern # 9: Overall Suitability: In summary, I find the article suitable for publication after making these revisions.

Author response: Thank you for pointing this out. The reviewer is correct, and we have revised it.

Author action: We have updated the relevant contents in the paper.

Reviewer#3, Concern # 1: Write the abstract more clear with some output values.

Author response: Thank you for pointing this out. The reviewer is correct, and we have revised it.

Author action: We updated the relevant contents in the paper as follows.

We used the multidimensional disaster data input unit for water level forecasting, including hydrology, meteorology, and geography, and 5 days of time windows for forecasting, The comprehensive reliability of LSTM-Bayesian for 0~102 hours flood reached 92.31%, and the comprehensive accuracy reached 89.15%, and confidence interval forecasting using LSTM is the best method, and achieved reasonable balance of reliability and accuracy. Overall, compound RNN could be a good alternative for forecasting hourly streamflow and extreme water level in small watersheds.

Reviewer#3, Concern # 2: Contribution is not clear.

Author response: Thank you for pointing this out. The reviewer is correct, and we have revised it.

Author action: We updated the relevant contents in the paper as follows.

The biggest contribution of the confidence interval forecasting model of small watered flood based on compound RNN and Bayesian proposed in this paper is that the RNNs-Bayesian models can consider both reliability and accuracy of flood water level prediction, which is also of great concern in hydrological forecasting. Of course, it should also be considered that the model relies on the high-quality near time series data of the monitoring points' prediction time points, and has excellent prediction effect for the short term (0~72h), but the reliability and accuracy of the medium and long-term prediction are not enough. The uncertain prediction model will play an important role in the prediction, early warning, preview and pre planning of small watershed.

Reviewer#3, Concern # 3: In the introduction and conclusion, emphasize the practical applications and real world impact of the proposed model to align the technical work with practical significance.

Author response: Thank you for pointing this out. The reviewer is correct, and we have revised it.

Author action: We updated the relevant contents in the paper as follows.

The error rate of flood warning is high, it is of great significance to study the prediction of flood in small watersheds to reduce the complex sudden flood disasters.

It improves the reliability of flood warning in small watersheds, and can provide reliable information for the decision-making of flood control and disaster management of government departments to avoid flood disasters.

Reviewer#3, Concern # 4: Analyse the performance of the algorithm as the prediction steps increases.

Author response: Thank you for pointing this out. The reviewer is correct, and we have revised it.

Author action: We updated the relevant contents in the paper as follows.

As the prediction steps increases, according to Figure 6, 7 and 8, the prediction confidence interval is getting larger and larger, but the ability to cover the real value is getting lower and lower, both reliability and accuracy tend to decline, a phenomenon that aligns with established principles regarding uncertainty. The reliability is not guaranteed, and the accuracy is getting lower and lower. In addition, according to Table 8, when the prediction window is the 5 days, the MDAD is the smallest. The bigger it is, the worse the comprehensive performance of reliability and accuracy is.

---

## [Editor Report · Decision Letter 2]

10 Mar 2025

Confidence interval forecasting model of small watershed flood based on compound Recurrent Neural Networks and Bayesian

PONE-D-24-42128R2

Dear Dr. Wang,

We’re pleased to inform you that your manuscript has been judged scientifically suitable for publication and will be formally accepted for publication once it meets all outstanding technical requirements.

Kind regards,

Jinran Wu, PhD

Academic Editor

PLOS ONE

---

## [Editor Report · Acceptance letter]

PONE-D-24-42128R2

PLOS ONE

Dear Dr. Wang,

I'm pleased to inform you that your manuscript has been deemed suitable for publication in PLOS ONE. Congratulations! Your manuscript is now being handed over to our production team.

Kind regards,

on behalf of

Dr. Jinran Wu

Academic Editor

PLOS ONE